# Bronchial Asthma and Sarcopenia: An Upcoming Potential Interaction

**DOI:** 10.3390/jpm12101556

**Published:** 2022-09-21

**Authors:** Nikolaos D. Karakousis, Ourania S. Kotsiou, Konstantinos I. Gourgoulianis

**Affiliations:** 1Primary Healthcare, Internal Medicine Department, Amarousion, 15125 Athens, Greece; 2Faculty of Nursing, University of Thessaly, 41500 Larissa, Greece; 3Department of Respiratory Medicine, Faculty of Medicine, University of Thessaly, 41110 Larissa, Greece

**Keywords:** asthma, sarcopenia, low muscle mass, inflammation, respiratory disease

## Abstract

Background: Sarcopenia seems to be an emerging health issue worldwide, concerning the progressive loss of skeletal muscle mass, accompanied by adverse outcomes. Asthma is a chronic inflammatory respiratory condition that is widespread in the world, affecting approximately 8% of adults. Although data are scarce, we aim to shed light on the potential association between low muscle mass and asthma and point out any probable negative feedback on each other. Methods: We searched within the PubMed, Scopus, MEDLINE, and Google Scholar databases. Study selections: Three studies were included in our analysis. Only original studies written in English were included, while the references of the research articles were thoroughly examined for more relevant studies. Moreover, animal model studies were excluded. Results: 2% to 17% of asthmatics had sarcopenia according to the existent literature. Sarcopenic asthmatic patients seem to have reduced lung function, while their mortality risk may be increased. Furthermore, patients with asthma- chronic obstructive pulmonary disease (COPD) overlap syndrome phenotype and sarcopenia might have a higher risk of osteopenia and osteoporosis progression, leading consequently to an increased risk of fractures and disability. Conclusions: Emerging data support that pulmonologists should be aware of the sarcopenia concept and be prepared to evaluate the existence of low muscle mass in their asthmatic patients.

## 1. Introduction

Worldwide, it there is an emerging interest concerning the progressive loss of skeletal muscle mass and loss of muscle function, broadly known as sarcopenia [1]. Sarcopenia prevalence in the elderly is considered quite variable, ranging from 5% to 50%, depending on different factors such as age, gender, pathological conditions, and last but not least, criteria concerning diagnosis [1]. Moreover, it is closely related to frailty syndrome, which is related to increased vulnerability [2]. Besides the aging process, low muscle mass can also be associated with pathological conditions. Among these conditions are chronic liver and kidney disease, inflammatory bowel disease, diabetic foot, and many others [2,3,4,5].

Asthma is a chronic inflammatory disorder concerning the airways [6]. It is characterized by chronic airway inflammation, which is manifested as variable airway narrowing leading to wheezes, dyspnea, and cough [7]. Asthma affected an estimated 262 million people in 2019 [1] and caused 455,000 deaths [8]. It seriously affects people’s physical along with their mental health, resulting in limited physical activity and decreased quality of life (QoL) [8].

In this non-systematic review, we aim to investigate the potential interplay between these two clinical entities, even though data are limited and further studies are needed to validate this interaction.

### 1.1. The Concept of Sarcopenia: Where We Stand?

The combination of low muscle mass and low muscle function is characterized as sarcopenia [9,10]. Even though this term was used to describe the loss of muscle mass and physical performance associated with aging, nowadays, factors harming sarcopenia progression may concern chronic diseases, an idle lifestyle, disability, and malnutrition [9,11]. It is already established that alterations in mitochondrial function, muscle fiber types, myokines, nicotinamide adenine dinucleotide (NAD+) metabolism, and gut microbiota are present in aged muscle compared to young muscle or healthy aged muscle [12]. Several age-related factors, such as neuromuscular degeneration, changes in hormone levels, chronic inflammation, and oxidative stress, are related to the development of low muscle mass [13]. On the other hand, low muscle mass might be related to pathological clinical conditions such as chronic kidney disease (CKD), chronic liver disease, respiratory disease, endocrine disorders, and others [4,14,15,16].

Sarcopenia is an important component of the syndrome of frailty, which is associated with increased vulnerability, a decline in the physiological reserves of several systems of the human body and augmented susceptibility to both endogenous and exogenous stressors [17,18]. Frailty syndrome has also been associated and linked to the aging population, and other pathological conditions such as postoperative complications, metabolic syndrome, cardiovascular disease, inflammation and many others [17,19,20].

In 2010, the European Working Group on Sarcopenia in Older People (EWGSOP) published a sarcopenia definition. Still, in early 2018, the Working Group carried out a new meeting (EWGSOP2) to determine an update concerning the description of this condition. In its 2018 definition, EWGSOP2 uses low muscle strength as the basic parameter of sarcopenia [21]. The updated consensus on sarcopenia uses detection of low muscle quantity and quality to confirm the sarcopenia diagnosis while it identifies poor physical performance as indicative of severe sarcopenia. Moreover, it updates the clinical algorithm used for sarcopenia case-finding, diagnosis and confirmation, and severity determination and provides clear cut-off points for measurements of variables that identify and characterize sarcopenia [21].

To monitor sarcopenia among individuals, there are specific tools. One screening tool for sarcopenia is SARC-F, which is a questionnaire consisting of five questions: Strength (S), Assistance walking (A), Rising from a chair (R), Climbing stairs (C), and Falls (F) on a scale of 0 to 2. The cutoff value recommended is ≥4 points [22,23]. In addition, tools such as grip strength and chair stand test (chair rise test), gait speed, timed-up-and-go test (TUG), 400-m walk or long-distance corridor walk (400-m walk) and short physical performance battery (SPPB) may also be of great importance to assess skeletal muscle strength and physical performance [21].

The laboratory evaluation of skeletal muscle mass, or skeletal muscle quality, can be carried out by appendicular skeletal muscle mass (ASMM) by Dual-energy X-ray absorptiometry (DXA), muscle ultrasonography, neutron activation (NAA), electrical impedance myography (EIM), whole-body skeletal muscle mass (SMM) or ASMM predicted by Bioelectrical impedance analysis (BIA) and lumbar muscle cross-sectional area by CT or MRI [21,24,25].

Interventions concerning sarcopenia are also critical to prevent its progression and adverse outcomes. Among these interventions are dietary supplementation, exercise interventions, and combined diet and exercise interventions or lifestyle interventions [26].

Both aerobic and resistance training seem to increase muscle strength and improve physical function in general [13]. Specifically, in the early 1990s, a series of studies established the role of Progressive Resistance Exercise Training (PRT) in increasing muscle size, muscle strength, and functional capacity in the elderly. At the same time, in 2009, a Cochrane review on 121 trials concluded that PRT could be imperative to improve physical performance along with muscle strength, including gait speed and getting up from a chair. PRT should be considered a first-line treatment strategy for managing and preventing sarcopenia and its adverse outcomes, but trained therapists and special equipment are required for its implementation [13].

It is already well-established that malnutrition is related to the pathogenesis of low muscle mass, specifically in frail and vulnerable elderly patients [13,18,27,28]. Interventions concerning nutrition may include increased protein, vitamin D supplementation, creatine monohydrate, antioxidants, omega-3 fatty acids, and other nutritional strategies, but all these are under consideration [13,18,29].

### 1.2. Bronchial Asthma: A Respiratory Key Competitor

Bronchial asthma is a medical condition that may have detrimental effects, while its prevalence globally has demonstrated a rapid increase during the last century [30,31]. It is a common clinical condition due to chronic inflammation of the lower respiratory tract, whilst due to the fact that it is a quite heterogenic clinical condition, it is often underdiagnosed, despite the fact that its clinical manifestation is already well-established and there are already valid and quite effective treatment strategies in order to confront this medical issue [32].

The risk factors concerning the bronchial asthma are already validated and it seems that gene-environment interactions have a pivotal role [32,33]. As has already been proven, genetics and heritability have an important role in bronchial asthma development along with epigenetic variation, whilst respiratory infections, particularly viral, are associated with environmental exposures, tobacco smoke, pollutants, ozone, atopic conditions, chemical exposures and effects of the microbiome, stress and metabolites [32,33,34,35].

The pathophysiology of this clinical issue is closely linked to the inflammation of the lower airway. This is most likely to derive from the combination of environmental exposures, genetic profile of its individual and probably alterations in the microbiome and metabolites [32,36]. It is well-established that the most frequent type of inflammation in asthmatic patients is the type 2 inflammation which can be associated with eosinophilic disorders, allergic diseases and parasite infections [32,37]. In addition, type 2 inflammation in asthmatic individuals can be characterized by increased IL-33 and thymic stromal lymphopoietin, increased OX40L expression and lymph node migration affecting lymphocyte maturation, metaplasia and increased mucin stores, increased TH2 bias with downregulation of Treg cells, along with increased IL-4, IL-5 and IL-13, increased IgE-producing plasma cells, IL-5–mediated accumulation and increased IgE binding and mediator storage [32,37]. All these alterations, which happen in the lower airways may lead to a remodeling status of the lung tissue in asthmatic subjects, in the mucosa and submucosa, including epithelial hyperplasia and metaplasia of goblet cells along with increased mucus production, smooth muscle hypertrophy, collagen deposition and larger mucous glands leading to airways remodeling and narrowing and increased mucous production [32,38,39]. Moreover, it is of great importance to point out, that in individuals with asthma, chest wall geometry is modified, shortening the inspiratory muscles and as a result the ability of these muscles to generate tension is quite reduced [40].

There are four essential symptoms concerning individuals living with bronchial asthma. Among these symptoms are: wheezing, coughing, shortness of breath / dyspnea and chest tightness [32,41], whilst the differential diagnosis includes medical conditions such as reactive airway disease, bronchopulmonary dysplasia, bronchiolitis and chronic obstructive pulmonary disease (COPD) [32,42].

Bronchial asthma classification concerns intermittent or persistent asthma, ranging from mild to severe, while certain asthmatic patients may have intermittent to persistent asthma [6]. Moreover, another classification concerning specific types of asthma causes and manifestations such as non-allergic, allergic, aspirin-exacerbated respiratory disease, occupational, potentially fatal, exercise-induced, and cough variant asthma [6].

Asthma management remains still quite intriguing, with acute asthma being a medical emergency that could be fatal [43]. In addition, it is well-known that asthma severity is characterized by the presence of exacerbations [43]. The four fundamental components of asthma management include patient education, monitoring and recording of symptoms and lung function, control of triggering factors and conditions that fuel comorbidity, requiring pharmacologic treatment administration [44,45,46].

Asthma treatment strategy is associated with the administration of inhaled corticosteroids (ICS), which have the ability to reduce asthma exacerbations and generally ameliorate the disease control [46]. In addition, in individuals living with this chronic respiratory disease, poorly controlled asthma and a history of prior asthma exacerbations, the combined administration of ICS and long-acting β-agonists (LABA), such as budesonide and formoterol, can lead to a significant reduction of asthma exacerbations compared to ICS administration alone, whilst the prescription of ICS/LABA combinations, both for maintenance and symptom relief, has demonstrated reduction concerning asthma exacerbations [46,47]. Regarding other treatment strategies, leukotriene antagonists seem to reduce exacerbations both in children and adults, while montelukast reduced asthma exacerbations to RV infections among children, even if adding montelukast to inhaled budesonide was as effective as doubling the dose of inhaled budesonide [46]. Last but not least, the administration of anticholinergic drugs, such as tiotropium, reduces the frequency of asthma exacerbations and is approved by Food and Drug Administration (FDA) for long-term, maintenance treatment for individuals 6 years of age and older with persistent asthma, uncontrolled with ICS along with the use of one or more drugs against bronchial asthma [46,48].

It is important to underline that, in severe conditions of bronchial asthma, there is availability of biologic therapies in the form of anti-IgE (omalizumab) and anti-IL5 therapies (mepolizumab and reslizumab) [49].

Asthma and obesity, both of which are considered global health issues, tend to increase in parallel indicating a potential link between these two conditions [50,51]. There is a debate whether body mass index (BMI) status is associated with asthma control, i.e., the persistence and intensity of symptoms of asthma. [50,51].

### 1.3. Literature Review Organization

In this non-systematic review article, the current literature was retrieved using the PubMed, Scopus, MEDLINE, and Google Scholar databases from the date of the idea’s inception concerning this review from July 1975 until August 2022. We have searched for the following terms: “sarcopenia and asthma” OR “sarcopenia and bronchial asthma” OR “low muscle mass and asthma” OR “low muscle mass and bronchial asthma”. Only original studies written in English were included, while the references of the research articles were thoroughly examined for relevant studies. Animal model studies were excluded. In this study, we tried to highlight the existing literature concerning the interaction between these two entities (Table 1).

### 1.4. Sarcopenia and Bronchial Asthma: The Intriguing Interplay

The association between sarcopenia and bronchial asthma seems to have been under medical investigation in recent years. Researchers worldwide tried to investigate the potential impact of these two entities on each other. Still, there is enough scientific space for a further and more thorough investigation.

Won et al. tried to investigate the association between sarcopenia and asthma in the elderly, mainly concerning asthma control and lung function [52]. The groups under investigation were divided and analyzed related to muscle mass, asthma, and physical activity. They have demonstrated that sarcopenic asthma had a younger onset and reduced physical activity than non-sarcopenic asthma, whilst asthma control was not associated with physical activity and low muscles mass [52]. Moreover, using multivariate logistic regression analyses, they further pointed out that sarcopenic asthma was associated with airway obstruction (FEV1 < 60%), older age, male gender, and lower body mass index (BMI), compared with non-sarcopenic asthma [52]. Their conclusions highlighted that intense physical activity and sarcopenia might contribute to reduced lung function in elderly asthmatics [52].

Benz et al. focused on investigating the association between sarcopenia, higher systemic immune-inflammation index (SII), COPD or asthma, and all-cause mortality in a large-scale population-based setting, taking under serious consideration that SII and sarcopenia are associated with higher morbidity in patients with COPD or asthma [53]. 4482 participants, aged > 55 years, with 57.3% being female, from the population-based Rotterdam Study were included. Asthma and COPD patients were diagnosed based on spirometry and clinical examination [53]. They defined sarcopenia according to the updated EWGSOP2 criteria while handgrip strength was obtained from the non-dominant hand using a hydraulic dynamometer, and appendicular lean mass was measured by DXA [53]. Independent of the presence of sarcopenia, COPD or asthma participants had a higher risk of all-cause mortality (HR: 2.13, 95% CI 1.46–3.12 and HR: 1.70, 95% CI 1.32–2.18 for those with and without sarcopenia, respectively, while higher SII levels increased mortality risk even in people without sarcopenia, COPD or asthma [53]. In conclusion, they pointed out that middle-aged and older people with COPD, higher SII levels, or sarcopenia had an independently increased mortality risk. At the same time, they recommended that sarcopenia and SII assessment in everyday medical practice could be predictors of worse progress in the elderly with COPD or asthma [53].

Lee et al. investigated the association between sarcopenia and bone mineral density (BMD) (which is related to osteopenia and osteoporosis) in asthma-COPD overlap (ACO), based on the existing hypothesis that sarcopenia and decreased BMD are common in the elderly and are significant comorbidities concerning obstructive airway disease (OAD) [54]. A total of 947 subjects were included in the study: 89 had asthma, 748 had COPD, and 110 ACO underwent qualified spirometry and DXA. This comparative study demonstrated that the sarcopenia group had higher risks of developing osteopenia, osteoporosis, and low BMD than the non-sarcopenia group in the ACO phenotype (OR: 6.620, 95% CI 1.129–38.828; OR: 9.611, 95% CI 1.133–81.544; and OR: 6.935, 95% CI 1.194–40.272, respectively), while in the asthma phenotype, the sarcopenia group showed no increased risk in comparison with the non-sarcopenia group [54]. They have concluded that in the ACO phenotype, individuals with sarcopenia had a higher prevalence rate and higher risks of osteopenia and osteoporosis than those without sarcopenia among all OAD phenotypes [54]. Osteoporosis is a significant factor in fractures and, as a result, disability, mortality, and morbidity [55,56]. It is already well established that the cost of osteoporosis adverse outcomes carries a significant economic burden concerning all countries, globally [57].

Figure 1 summarizes the explain the relationship between bronchial asthma and sarcopenia.

## 2. Discussion

This non-systematic review aims to demonstrate and highlight the potential interplay between bronchial asthma and low muscle mass, known as sarcopenia. These two entities play a pivotal role in respiratory and muscle health, respectively, while they are already linked to many other pathological conditions and adverse outcomes that could deteriorate the QoL among individuals. However, the existing literature is still scarce but quite promising.

Certain limitations are related to this medical issue. These limitations are associated with the currently small number of studies investigating this intriguing interplay, while the number of patients participating in them is relatively limited.

Nevertheless, it seems that it would be intriguing if further studies could include and investigate a more significant number of patients living with bronchial asthma, not only older but also of younger age, and evaluate the existence or not of a low muscle state by muscle mass assessment. In addition, it is important to study the effect of currently used treatment against asthma in sarcopenic individuals living with asthma and whether these agents could positively impact muscle mass, apart from their chronic respiratory disease.

Another interesting approach concerning this medical topic and the potential interplay between these entities could be the development of specific indexes that could evaluate the prognosis of bronchial asthma among asthmatic sarcopenic patients, probably relying on their clinical image, along with laboratory parameters concerning both muscle mass and respiratory activity.

In addition, using specific biomarkers that could assess sarcopenia phenotype in asthmatic subjects might be of great importance. It has already been analyzed the association with plasma biomarkers such as glycoprotein Dickkopf-3 (Dkk-3), c-terminal agrin fragment-22 (CAF22), and microRNAs miR-21, miR-134a, miR-133 and miR-206 with hand-grip strength (HGS) and appendicular skeletal mass index (ASMI) in male, 54–73-year-old individuals with COPD, bronchial asthma or pulmonary tuberculosis and it has been demonstrated a modest-to-significant increase in the plasma markers of oxidative stress, inflammation and muscle damage, which had varying degrees of correlations with Dkk-3, CAF22 and selected micro RNAs (miRs) in these respiratory diseases [58]. This could imply that these biomarkers could be significant and valuable tools to evaluate the phenotype of sarcopenia among older patients with diseases concerning their respiratory system [58].

Last but not least, it would be appealing if an interventional protocol could be established for sarcopenic individuals with bronchial asthma. This specific protocol could include a multimodal approach in which nutrition, exercise, and respiratory rehabilitation programs could beneficially coexist and positively affect the muscle mass, along with the asthmatic exacerbations. Ameliorating these conditions could have an upside effect on these individuals and improve their QoL.

## 3. Conclusions and Future Perspectives

Sarcopenic patients living with a chronic respiratory disease, such as bronchial asthma, may have reduced lung function, while their mortality risk may increase. In addition, individuals with asthma-COPD overlap syndrome phenotype and low muscle mass may have a higher risk of osteopenia and osteoporosis progression, leading consequently to an increased risk of fractures, immobilization, and disability. Pulmonologists should be aware of the sarcopenia clinical condition and be prepared to evaluate low muscle mass in bronchial asthma patients using the existing screening tools for sarcopenia. Moreover, physicians who examine sarcopenic patients with bronchial asthma should be able to appropriately collaborate with specialists who deal with nutrition and exercise, giving their patients a multimodal approach concerning these entities’ interplay and the optimum treatment.

## Figures and Tables

**Figure 1 jpm-12-01556-f001:**
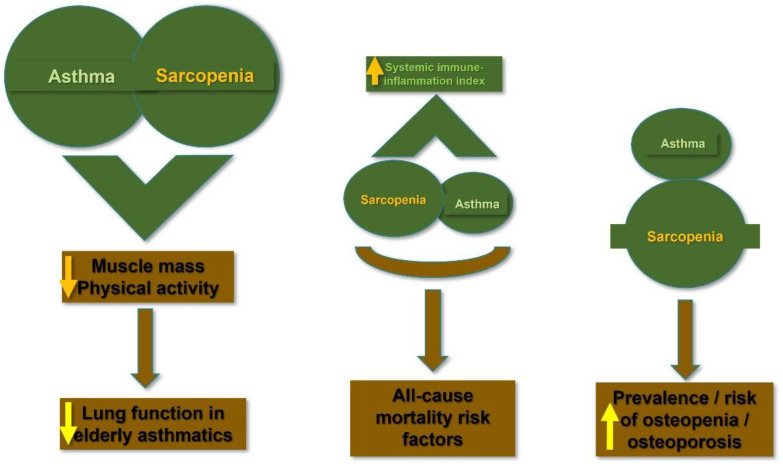
The relationship between bronchial asthma and sarcopenia.

**Table 1 jpm-12-01556-t001:** The interplay between sarcopenia and bronchial asthma.

Authors ^Ref^/Year	Study Design	Study Population	Findings	% of Sarcopenia in Asthmatics	Low Muscle Mass Evaluation
Won et al. [50]/2022	Cross-sectional	320 elderly asthmatics	Decreased muscle mass and physical activity levels may contribute to reduced lung function concerning elderly asthmatics. Sarcopenic asthma was associated with low BMI, aging, reduced lung function.	15% asthmatics has sarcopenia	Appendicular skeletal muscle was calculated as the sum of the skeletal muscle mass.
Benz et al. [51]/2022	Population-based study	4482 participants (aged > 55 years; 57.3% female) from the population-based Rotterdam Study	Middle-aged and older people with COPD, higher SII levels or sarcopenia had an independently increased mortality risk, whilst routinely evaluating sarcopenia and SII in older people with COPD or asthma is recommended	1.4% asthmatics with sarcopenia, 2.1% COPD patients with sarcopenia	Handgrip strength evaluated by hydraulic dynamometer and appendicular lean mass measured by DXA
Lee et al. [52]/2017	Comparative study	947 subjects were included in the study: 89 had asthma, 748 COPD, and 110 ACOS	In the ACO phenotype, sarcopenic individuals had a higher prevalence rate and risks of osteopenia and osteoporosis than those non-sarcopenic	17.1% asthmatics with sarcopenia, 50.5% COPD patients with sarcopenia	Assessment by DXA

Abbreviations: ACO, Asthma—COPD overlap, COPD, chronic obstructive pulmonary disease; DXA, Dual-energy X-ray overlap.

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
