# Peer review of "Bronchial Asthma and Sarcopenia: An Upcoming Potential Interaction"

_jpm, 2022, doi:10.3390/jpm12101556_

Round 1

Reviewer 1 Report

The article is deal with the interaction of bronchial asthma and sarcopenia. The topic discussed is very important for the treatment and prevention of asthma and sarcopenia.

I would like to make a few comments:

1)            line 11: “that is the commonest in the United States”

            There is no confirmation in the text that asthma is the commonest in the United States .  It is better to use another phrase, for example “widespread in the world”

2)            line 28: and further in the text:

          Use square brackets for cited literature

3)            line 39: “250,000 people have lost their lives due to broncial asthma”

Asthma affected an estimated 262 million people in 2019 (1) and caused 455 000 deaths:

https://www.who.int/news-room/fact-sheets/detail/asthma

4)         line 39: broncial –the word has mistake

bronchial – correct

5)      line 118: add to the citation [32, 33] recent articles:

Ntontsi, P.; Photiades, A.; Zervas, E.; Xanthou, G.; Samitas, K. Genetics and Epigenetics in Asthma. Int. J. Mol. Sci. 2021, 22, 2412. https://doi.org/10.3390/ijms22052412

Guryanova, S.V.; Gigani, O.B.; Gudima, G.O.; Kataeva, A.M.; Kolesnikova, N.V. Dual Effect of Low-Molecular-Weight Bioregulators of Bacterial Origin in Experimental Model of Asthma. Life 2022, 12, 192. https://doi.org/10.3390/life12020192

6)  line 187: The table mentioned in the text is missing.

It is necessary to describe in the table for every investigation: the number of patients, their age, number  of male and female, and other available data. It is necessary to give the % of sarcopenia among suffering from bronchial asthma.

7) Add to the ABSTRACT the % of  sarcopenia among suffering from bronchial asthma.

Author Response

Response to Reviewer 1 Comments:

  1. The article is deal with the interaction of bronchial asthma and sarcopenia. The topic discussed is very important for the treatment and prevention of asthma and sarcopenia.

RESPONSE : We sincerely thank you for your kind words about our paper. We are delighted to receive a positive feedback from you.

        I would like to make a few comments:

  • line 11: “that is the commonest in the United States”

 There is no confirmation in the text that asthma is the commonest in the United States .  It is better to use another phrase, for example “widespread in the world” 

RESPONSE : We really thank you for this point. In the revision, we have revised it.  

  1. line 28: and further in the text: Use square brackets for cited literature

RESPONSE : Thank you for this point. In the revised manuscript, we have used square brackets for cited literature.

  1. line 39: “250,000 people have lost their lives due to broncial asthma”

Asthma affected an estimated 262 million people in 2019 (1) and caused 455 000 deaths: https://www.who.int/news-room/fact-sheets/detail/asthma

RESPONSE : Thank you for this comment. In the revision, we have replaced the mortality data.

  1. line 39: broncial –the word has mistake

bronchial – correct

RESPONSE : Thank you for thispoint and we apologize for this typographical error.

  1. line 118: add to the citation [32, 33] recent articles:

Ntontsi, P.; Photiades, A.; Zervas, E.; Xanthou, G.; Samitas, K. Genetics and Epigenetics in Asthma. Int. J. Mol. Sci. 2021, 22, 2412. https://doi.org/10.3390/ijms22052412

Guryanova, S.V.; Gigani, O.B.; Gudima, G.O.; Kataeva, A.M.; Kolesnikova, N.V. Dual Effect of Low-Molecular-Weight Bioregulators of Bacterial Origin in Experimental Model of Asthma. Life 2022, 12, 192. https://doi.org/10.3390/life12020192

RESPONSE : Thank you for the comment. The citations have been added, as suggested.

  1. line 187: The table mentioned in the text is missing. It is necessary to describe in the table for every investigation: the number of patients, their age, number of male and female, and other available data. It is necessary to give the % of sarcopenia among suffering from bronchial asthma.

RESPONSE : Thank you for this comment. In the revised manuscript, we have added the Table 1. 

  1. Add to the ABSTRACT the % of sarcopenia among suffering from bronchial asthma.

RESPONSE: Thank you for this comment. In the revision we have added the % of sarcopenia among suffering from bronchial asthma. 

We appreciate you taking the time to offer us your insights related to the paper. We found your feedback very constructive. We tried to be responsive to your concerns.

Reviewer 2 Report

The review is well-articulated, but I suggest adding at least one figure to explain the relationship between the mechanism of bronchial asthma and sarcopenia. Also, if possible,  include a table to clearly show the numbers in a tabular form. 

Author Response

Response to Reviewer 2 Comments: 

  1. The review is well-articulated, but I suggest adding at least one figure to explain the relationship between the mechanism of bronchial asthma and sarcopenia. Also, if possible, include a table to clearly show the numbers in a tabular form.

RESPONSE: We sincerely thank you for your kind words about our paper. We are delighted to receive a positive feedback from you. Thank you for this comment. In the revised manuscript, we have added Figure 1 to explain the relationship between bronchial asthma and sarcopenia, as well as a table to clearly show the numbers in a tabular form.

We appreciate you taking the time to offer us your insights related to the paper. We hope you find these revisions rise to your expectations.

Round 2

Reviewer 1 Report

In the table 1 there is a mistake:

1) 15% athmatics should be corrected: 15% asthmatics;

2) In footnote give the explanation of ACOS.

Author Response

RESPONSE TO REVIEWER 1:

In the table 1 there is a mistake:

1) 15% athmatics should be corrected: 15% asthmatics;

2) In footnote give the explanation of ACOS.

RESPONSE: Thank you for the comments, and we apologize for the error made by omission. We have revised Table 1, as suggested.

We found your feedback very constructive. We tried to be responsive to your concerns. We really thank you for taking the time and energy to help us improve this paper